# Microstructure and Corrosion Resistance of AA4047/AA7075 Transition Zone Formed Using Electron Beam Wire-Feed Additive Manufacturing

**DOI:** 10.3390/ma14226931

**Published:** 2021-11-16

**Authors:** Andrey Filippov, Veronika Utyaganova, Nikolay Shamarin, Andrey Vorontsov, Nikolay Savchenko, Denis Gurianov, Andrey Chumaevskii, Valery Rubtsov, Evgeny Kolubaev, Sergei Tarasov

**Affiliations:** Institute of Strength Physics and Material Science, Siberian Branch of Russian Academy of Sciences, Akademicheskiy Pr., 634055 Tomsk, Russia; avf@ispms.ru (A.F.); shnn@ispms.ru (N.S.); vorontsov.a.583@gmail.com (A.V.); savnick@ispms.tsc.ru (N.S.); desa-93@mail.ru (D.G.); tch7av@gmail.com (A.C.); rvy@ispms.tsc.ru (V.R.); eak@ispms.ru (E.K.)

**Keywords:** additive manufacturing, aluminum alloy, functionally graded material, corrosion

## Abstract

A gradient transition zone was obtained using electron beam deposition from AA4047 wire on AA7075 substrate and characterized for microstructures, tensile strength and corrosion resistance. The microstructure of the transition zone was composed of aluminum alloy grains, Al/Si eutectics and Fe-rich and Si-rich particles. Such a microstructure provided strength comparable to that of AA7075-T42 substrate but more intense corrosion due to the higher amount of anodic Mg_2_Si particles. The as-deposited AA4047 zone formed above the transition zone was composed of aluminum alloy dendrites and interdendritic Al/Si eutectics with low mechanical strength and high corrosion potential.

## 1. Introduction

Functionally graded materials (FGMs) constitute a class of modern materials that allow the combination of unique physical, mechanical and chemical characteristics inherent in dissimilar materials into a single application or product. FGMs from aluminum alloys are fabricated with the use of various processes, such as gravitational, centrifugal or magnetic field-assisted casting [1,2], spark plasma sintering and infiltration [3]. Microstructures and defects formed in a transition zone between two dissimilar alloys are of special importance from the viewpoint of mechanical strength and corrosion. Therefore, great attention must be paid to studying and improving transition zone characteristics.

For instance, gravitational casting methods allowed the production of transition zones characterized by the presence of oxide and slug inclusions as shown by the example of AlSi7Mg0.3/AlSi12CuNiMg FGM [4]. Therefore, more effective casting methods were used, such as the centrifugal casting of such alloys as AA356, AA413 and AA390 to achieve the effect of grading the content of Si from hypoeutectic to hypereutectic [5]. Moreover, AA2014-SiC centrifugal cast and aged FGMs were obtained that allowed an improvement in their hardness and resistance to fatigue as compared to their standard cast states [6]. The wear resistance and strength characteristics of AA6061 and AA6061/SiC FGMs were also improved by centrifugal casting [7]. Al-Zn, Al-Ni and Al-Cu FGMs were obtained via directional solidification in a static magnetic field [8]. High-strength Al3003/Al6063 FGM was successfully fabricated using hot extrusion for multifunctional applications [9].

However, the use of the traditional above-described methods for fabricating FGMs does not allow the fabrication of complex-shaped void- and crack-free components with the desired composition gradient. Another drawback is a necessity to allow for extra thickness for further machining.

Additive manufacturing is a fast-developing technology that is being increasingly more widely applied to mass-scale production, in particular, for direct energy deposition (DED) from wire by means of arc or plasma transferred arc additive manufacturing (WAAM) or electron beam additive manufacturing (EBAM).

DED methods possess a number of well-known advantages over other additive methods, such as selective laser melting (SLM) and spark plasma sintering (SPS). The DED process allows in situ control of the wire (wires) feed along with other process parameters, thus growing a near-net shape article even from dissimilar materials [10]. The layer-by-layer DED fabrication of FGMs is very difficult work not only from the viewpoint of selecting the appropriate process and adjusting its parameters but also due to the differences of the physical and chemical characteristics of the dissimilar materials used. The problems resulting from these differences in DED may even be unavoidable and insuperable from the viewpoint of obtaining functional, resilient and defectless FGMs [11].

WAAM is a rather simple process that provides a very high deposition rate, which, however, does not guarantee the absence of defects, such as pores, especially when building up thin walls from aluminum alloys [12,13,14]. The WAAM fabrication of thin-walled components from AA5356 and AA4047 was studied to show that both arc discharge length and pulse energy may have an effect on deposition accuracy [15]. SLM allows one to obtain good quality components from aluminum alloys [16,17,18], but obtaining zero porosity is a problem with this method even when using such alloy as AlSi10Mg [19].

The quality (and therefore the high costs of source powders) is another limitation of this method. Nevertheless, a Ti6Al4V/Al12Si FGM with brittle TiSi_2_ and Ti_3_Al intermetallic compounds (IMCs) in the transition zone was obtained using the laser engineered net shaping (LENS) method [20]. For example, an AlSi10Mg alloy was SLM deposited on an Al-Cu-Ni-Fe-Mg substrate with good adhesion [21]. A hybrid approach was proposed by Dolev et al., which allowed the expansion of the nomenclature of SLM-made Ti-6Al-4V high-strength articles [22].

EBAM allows the fine controlling and in situ monitoring of the process parameters, and, therefore, it is widely used for the growing variety of materials and components [23,24,25,26,27,28]. The application of EBAM on aluminum alloys is limited because of the evaporation of alloying elements, such as Mg and Zn, which is the function of heat input. Nevertheless, there are examples of using EBAM for the fabrication of dense components from aluminum alloys [29,30,31,32], including those of the 7XXX and 2XXX series, which were inclined to hot cracking [33].

EBAM was successfully applied for the fabrication of defectless Cu-AISI 304 stainless steel FGMs by feeding two wires simultaneously [34,35]. A hybrid powder bed/wire EBAM allowed a fully dense CuAl-B_4_C FGM to be obtained [36]. Another attempt was devoted to growing an AA5356 wall on an AA7075 substrate [37] in order to improve the corrosion resistance of the latter. Admixing Mg into the AA7075/AA5356 transition zone proved to be detrimental for its corrosion resistance due to the precipitation of anodic Al–Cu–Zn–Mg and Mg_2_Si particles. The as-deposited AA5356 zone was characterized by pure pitting corrosion due to the presence of anodic Mg_2_Si particles, which suffer Mg dealloying during corrosion and impair its corrosion resistance. Using a Mg-free aluminum alloy to form a transition zone on the AA7075 substrate would be interesting from the considerations mentioned above. Si is another component of Mg_2_Si detrimental particles, and, therefore, the effect of admixing it with a low Mg AA7075 metal in the transition zone has to be analyzed.

Aluminum 4XXX alloys contain silicon that attains high fluidity to the melt, provides low shrinking in solidification and good weldability. In addition, the good corrosion resistance of alloys such as AA4047 allows its use for the fabrication of cast engine blocks and wheels. Therefore, it could be worthwhile to fabricate a component that will combine the useful characteristics of alloys such as AA7075 and AA4047, i.e., high mechanical strength and corrosion resistance, respectively. It is unavoidable that some transition zone will form between these two parts whose mechanical and corrosion characteristics should be studied as dependent on the process parameters. The main characteristic of the EBAM that determines structural evolution in the transition zone is the heat input. It is desirable to adjust the heat input depending on the heat removal conditions and solidification during layer-by-layer deposition. Such an approach has previously been used on the AA7075/AA5356 system [37].

The objective of this work is to study the microstructures, mechanical characteristics and corrosion resistance of a transition zone formed by EBAM from an AA4047 wire on an AA7075 substrate.

## 2. Materials and Methods

Thin-wall samples (Figure 1c) were fabricated using layer-by-layer electron beam melting and deposition of AA4047 (Table 1) Ø 1.2 mm wire on an AA7075-T42 (hereinafter AA7075) (Table 2) 19.5 mm in height and 5 mm in thickness substrate. Residual pressure in the chamber was −5·10^−5^ Pa. Deposition rate, wire feed rate, beam current and accelerating voltage were 380 mm/min, 1344 mm/min, 20 μA and 30 kV, respectively. The wall dimensions (with substrate) were 95, 39 and 5 mm. The deposition strategy is shown in Figure 1b.

Samples were cut off from the printed wall and then characterized for microstructures, tensile strength, microhardness and corrosion resistance (Figure 1a). Samples for metallographic studies were prepared using mechanical grinding, polishing with the use of diamond pastes and etching in the Keller reagent (2 mL HF (48%); 3 mL HCl; 5 mL HNO_3_; 190 mL H_2_O) for 2 min. Optical confocal microscope Olympus (Olympus Scientific Solutions Americas, Waltham, MA United States), scanning electron microscope LEO EVO 50 (ZEISS, Oberkochen, Germany) attached with an EDS add-on, and transmission electron microscope JEOL JEM-2100 (JEOL Ltd., Tokyo, Japan) were used to study the microstructures and corroded surfaces of the samples. Phase composition was identified using an XRD instrument, DRON-7 (Burevestnik, Saint Petersburg, Russia), CoKa radiation, λ = 17,902 Å, operating at 36 kV and 22 mA with a scan range of 15–165° (2θ), step size of 0.05° and counting time of 17 s.

Microhardness was measured using a «Duramin-5» (Struers A/S, Ballerup, Danemark) microhardness tester at 50 N load. Tensile machine Testsystem 110M-10 (Testsystems, Ivanovo, Russia) was used for the determination of strength characteristics.

Corrosion testing in 3.5% NaCl was carried out using a potentiodynamic method and an Electrochemical Instruments P-45X (Electrochemical Instruments, Chernogolovka, Russia) potentiostat. The samples that were cut off according to the scheme in Figure 1a were used as working electrodes, while the reference one was Ag/AgCl with 3.5 M solution of KCl. A counter-electrode was a graphite rod.

## 3. Results

### 3.1. Macro and Microstructures

Samples for metallographic examination were machined from the EBAM built wall according to the scheme shown in Figure 1. A cross-section view allowed us to divide the built-up metal into three zones (Figure 2b), namely, top layer zone I, mid-height transition zone II and substrate zone III, which were represented by almost pure deposited AA4047, intermixed AA/4047/AA7075 and heat-affected AA7075, respectively.

#### 3.1.1. Zone I

Typical hypoeutectic as-cast structures represented by α-Al dendrites and Al/Si eutectics vertically grown along the building direction can be observed above the transition zone line (Figure 2a). A certain amount of both elongated and equiaxed particles can be found in this zone (Figure 3a,b). These particles are later identified as Al-Fe-Si ones using XRD and TEM.

#### 3.1.2. Zone II

The AA7075/AA4047 transition zone is in the form of a ~150-μm-thick layer composed of α-Al grains with Al/Si eutectics, formed due to the intermixing of both alloys in the melted pool, as well as coarse particles (Figure 2b, enlarged views, Figure 3a,d). Solidification of the melted pool in the transition zone starts from growing the Si-enriched α-Al dendrites from partially melted AA7075 grains. Excess Si is forced out to the interdendritic spaces where the Al/Si eutectics are then formed. The α-Al dendrites (Figure 3d, position 1–3) are characterized by the presence of <1 at% concentrations of Mg, Si, Cu and Zn (Figure 4) and also contain small Si crystallites precipitated there from the already solidified Al (Figure 2b).

The Al/Si eutectics are located at the α-Al grain boundaries and in the particle agglomerates (Figure 3d, position 4–6), where the concentration of Si becomes higher than 30 at% (Figure 4b). Another sort of particle, namely, grain boundary Si-Mg-Cu-Zn ones, with Si concentration <10 at% (Figure 4c), are observed in Figure 3d, position 7–8. Even less Si is detected in the grain boundary particle in Figure 3d, position 9–10.

#### 3.1.3. Zone III

The microstructure of an AA7075 substrate is represented by coarse 100 ± 25 μm α-Al grains (Figure 2b). Coarse intermetallic particles are observed both inside the grains and on their boundaries (Figure 3a,c). These particles were identified as η-MgZn_2_, θ-Al_2_Cu and S-Al_2_CuMg phases, which are commonly inherent in the AA7075 [37].

### 3.2. Element Concentration Profiles and Phases in Transition Zone II

EDS profiles for basic alloy elements as distributed across the transition zone show that the atomic concentration of magnesium, zinc and copper sharply reduce in the middle of the transition zone, while that of silicon sharply grows (Figure 5). It can also be noticed that the concentration of Mg becomes slightly higher starting from a wall height of 10 mm; i.e., up to 2.5 at/% of Mg was found in the as-deposited α-Al +Al/Si eutectic layers.

The XRD patterns that were obtained from the transition zone (Figure 6) allowed for the identification of the presence of phases, such as α-Al, Si and MgZn_2_, which formed from intermixed AA7075/AA4047 metals (Figure 6). The MgZn_2_ phase was detected with the AA7075 substrate along with the α-Al.

Some amount of Si was detected here, which descended from the above as-deposited AA4047 fully composed of α-Al and Si phases. Despite the fact that the top part of the wall contains up to 2.5 at% of Mg, no Mg-containing phases were detected there using the XRD. Some slight peaks were found in the as-deposited AA4047, which could be related to Fe-Si-Al phases.

The α-Al lattice parameter is changed along the wall height as shown in Figure 7 and in accordance with Zn, Cu and Mg concentration profiles (Figure 5).

### 3.3. TEM Study of Particles

A TEM examination of particles in the AA7075 heat-affected zone has previously been carried out after the layer-by-layer deposition of AA5356 [37]. Therefore, it could be of more interest now to examine AA4047/AA7075 transition zone II and AA4047 deposition zone I.

#### 3.3.1. Transition Zone

Several types of particles were detected in the thin foils cut off from the transition zone metal. According to EDS data, the particle (Figure 8, Table 3) located in the triple junction of α-Al grains is composed of ν-phase Mg-Zn-Cu-(Al) [38] and S-phase (Al_2_CuMg). Fine precipitates (Figure 8, position 7–9) are composed of Al, Zn, Mg and Cu (Table 3) with a high concentration of aluminum that does not allow their more accurate identification. 

Moreover, coarse >2 μm particles (Figure 9) that contain Al, Fe, Mn, Cu and Zn (Table 4) can be found, whose precipitation from the melt is provided by the presence of these elements with AA7075. These particles have been identified as Fe-rich ones elsewhere [37]. Secondary Al-Zn-Mg-Cu particles (Figure 9, position 3–7) are located along the grain boundaries. Their accurate identification is difficult due to their size and, therefore, the detection of more aluminum in the matrix (Table 4, position 3–7). The rod-like (Figure 9, Table 4, position 8 and 9) and needle-like (Figure 9, Table 4, position 10) precipitates may be η-phase (MgZn_2_) and S-phase (Al_2_CuMg).

The third sort of particles are compound core–shell Mg_2_Si/ν-phase ones (Figure 10, Table 5, position 1–4 and 5–6), which can precipitate from the melt at high solidification rates and the insufficiently fast diffusion of AA7075 components [38]. The rod-like precipitates can be S-phase (Al_2_CuMg) (Table 5, position 7) and η-phase (MgZn_2_) (Table 5, position 8).

#### 3.3.2. AA4047 As-Deposited Zone I 

It was shown above (Figure 2a) that the microstructure of the as-deposited AA4047 zone I is composed of α-Al dendrites and interdendritic Al/Si eutectics. TEM images of the thin foils cut off from this zone also allow the observation of small <6 μm α-Al grains with a necklace of Al/Si eutectics (Figure 11a). Fine 300–700 nm Si particles were detected both in the interdendritic spaces (Figure 11b,d) and inside the α-Al grains (Figure 11c).

Iron-rich Al-Fe-Si particles (Figure 12a), with their chemical composition presented in Figure 12b–d, were detected in this zone. According to the XRD results, in Figure 6, these particles might have been identified as Fe_2_Al_3_Si_3_ and Fe_2_Al_2_Si_3_, while their EDS compositions showed them close to that of Fe_2_Al_8_Si [39,40] plausibly because of the limited EDS probe accuracy and contamination with aluminum.

### 3.4. Mechanical Characteristics

Dog bone-shaped samples for tensile testing were cut off from the additive-deposited wall so that the transfer zone II would be in the middle of the gauge length. Stress–strain curves obtained from the test were of the type inherent in the ductile materials (Figure 13a) and allowed for the determination of the strength characteristics as follows: mean yield stress limit 122.7 ± 15.2 MPa, mean ultimate stress limit 180.5 ± 2.5 MPa and mean elongation to failure 7.9 ± 1.3%.

The microhardness profile was obtained along the wall height to reveal the high hardness of the AA7075 heat-affected zone and the bottom part of the rather narrow transition zone followed by a sharp fall in its top half to the numbers typical of the deposited AA4047 (Figure 13b). The microhardness number distribution coincides with that of the AA7075 alloying components, such as Mg, Zn and Cu (Figure 5).

It was observed that a fracture occurred within the as-deposited AA4047 zone I (Figure 14a,b) characterized by the presence of α-Al dendrites and interdendritic Al/Si eutectics (see enlarged insets to Figure 14b). Such a result was not a surprise if taking into account the differences in the strength and structure of the AA7075, transition zone and as-deposited AA4047. 

The fracture occurred in the as-deposited AA4047 zone where there were no components other than Al-dendrites, excess Si-crystallites and Al/Si eutectics. It could be that Si-crystallites (shown inside the yellow circles in Figure 15a,b) served as stress concentrators, but this as-cast structure was heterogeneous enough by itself. The fracture surface is characterized by the presence of dimples, ridges and ~3 μm cells (Figure 15a,b). Despite the fact that no deformation neck formed, the fracture may be related to the viscous type.

### 3.5. Corrosion

Potentiodynamic corrosion testing was carried on samples cut off from all three zones, namely, as-deposited AA4047 zone I, transition zone II and heat-affected AA7075 substrate zone III. The minimum value of the corrosion potential Ecorr was observed for a sample from transition zone II (Figure 16, Table 6), which is only 29 and 215 mV less than that of AA7075 substrate zone III and as-deposited AA4047 zone I, respectively.

Let us note that the mean corrosion potential behavior is reversed as compared to that of the corrosion current density; i.e., the maximum current density corresponds to the minimum corrosion potential and vice versa (Table 6). Hence, the transition zone demonstrated minimum corrosion potential and maximum corrosion current density, while as-deposited AA4047 is characterized by the maximum corrosion potential and minimum current density.

An anodic branch of the as-deposited AA4047 polarization curve is characterized by the presence of oscillations, which can be a symptom of metastable pitting; this starts at −518 mV and ends at −366 mV. It was not possible to measure the pitting potential on the transition zone metal because of the intense dissolution of the numerous intermetallic particles that started just after reaching the corresponding E_corr_ value; i.e., no passivation behavior was observed. A pitting potential value for the heat-affected AA7075 substrate was −678 mV.

The samples subjected to corrosion tests were then washed, dried and examined using a confocal optical microscope for surface corrosion damages. The surface of as-deposited AA4047 zone I can be described by the presence of isolated shallow and ~10-μm-sized pits (Figure 17a,c) without any noticeable corrosion-affected areas (Figure 17a).

No changes were observed in zone I or as it approached closer to transition zone II (Figure 17a,b and Figure 18b).

Notable corrosion occurred on the surface of the transition zone (Figure 17a,d). Presumably, corrosion in this zone developed both by grain boundaries and particle agglomerates similar to those observed in Figure 3d. Deep pits of up to 11 μm in depth formed there as a result of corrosion (Figure 18c). Less deep (<5 μm) and smaller area pits were also observed in the heat-affected AA7075, just below the transition zone (Figure 17a,e and Figure 18d). Almost the same type of corrosion surface pattern with ~6-μm-deep corrosion pits was observed in the heat-affected zone of the AA7075 substrate (Figure 17f). These above-described results allow the suggestion that a strong correlation exists between the structural state and the corrosion behavior on the samples investigated.

## 4. Discussion

The melting and transferring of AA4047 wire to the melt pool formed on an AA7075 substrate by an electron beam allowed us to obtain a narrow transition zone between two alloys where components of both alloys admixed and solidified in the form of aluminum alloy dendrites, interdendritic spaces filled with Al/Si eutectics and coarse intermetallic particles (agglomerates). According to the EDS profiles in Figure 5, high-density elements, such as Zn and Cu, stay in the bottom part of the transition zone as well as lighter-than-Al magnesium despite the fact that its somewhat increased concentration was detected in the top part of the wall. Such a finding may be caused by some floating of the magnesium in partial layer-by-layer remelting. The distribution of silicon across the transition zone also shows a sharp threshold between the bottom and top parts of the zone.

Therefore, a sharp concentration threshold exists instead of a somewhat wide transition zone. Such a finding can only be explained by the lack of intense intermixing in the melted pool. Several factors determine the intensity of metal intermixing in the pool, such as the intensity of heating, convection and density differences. The intensity of convection is dependent on the heat input, which has to be limited to also avoid the boiling and loss of such alloying elements as Mg and Zn.

The phase composition of the transition zone is determined by the distribution of alloying elements and impurities. The magnesium-, copper- and zinc-rich bottom part contains all particles inherent in the AA7075 substrate, while the top part shows more Si in the form of the core–shell Mg_2_Si/ν-phase particles. The as-deposited AA-4047 is composed of aluminum dendrites and Al/Si eutectics, thus representing as-cast AA4047 structures without any reinforcing fine and coherent precipitates. The microhardness test showed its hardness at the level of 0.6 GPa as compared to 1.25 and 1.3 GPa of the AA-7075 heat-affected zone and bottom part of the transition zone, respectively. Therefore, the tensile test results showed that fracture occurred in this zone with the minimum strength as compared to those of AA7075 substrate and transition zone.

It was shown that the transition zone was enriched by Si and, therefore, more Mg_2_Si precipitated there, thus causing the most intense corrosion. It is also known [41] that both Mg- and Si-containing particles, such as S-phase (Al_2_CuMg) and Mg_2_Si, serve as anodic ones with respect to their embedding in the α-Al matrix. Therefore, these particles are dissolved in NaCl and leave the deep pits (Figure 18c). The AA7075 substrate contains less Si, and, therefore, there are less Mg_2_Si particles to be subjected to corrosion.

On the contrary, cathodic Si and Fe-Si particles reveal high resistance to corrosion but provoke preferential and uniform dissolution of the matrix [42] instead of the formation of deep pits in the as-deposited AA4047 zone.

## 5. Conclusions

The additive electron beam wire-feed deposition of AA4047 on a high-strength AA7075 substrate resulted in the formation of a narrow defectless and resilient transition zone characterized by a sharp threshold in the distribution of alloying elements and, correspondingly, the inhomogeneous distribution of intermetallic phases. Such a transition zone is a result of using in situ adjustment of the heat input, which allowed us to minimize the intermixing of the elements in the melted pool and obtain relatively “cold” deposition conditions.

The minimum and maximum values of a corrosion potential were obtained on the surfaces of the as-deposited AA4047 and AA7075/AA4047 transition zone, respectively. The precipitation of anodic Mg_2_Si and Al_2_CuMg in the transition zone caused their intense dissolution and pitting.

## Figures and Tables

**Figure 1 materials-14-06931-f001:**
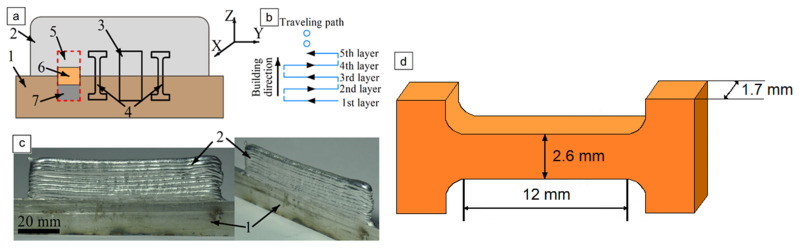
(**a**) Sample cut-off scheme; (**b**) deposition strategy; (**c**) additive-manufactured wall; (**d**) tensile sample with dimensions 1—AA7075 substrate; 2—as-deposited AA4047; 3—metallographic specimen; 4—tensile sample; 5–7—corrosion test samples.

**Figure 2 materials-14-06931-f002:**
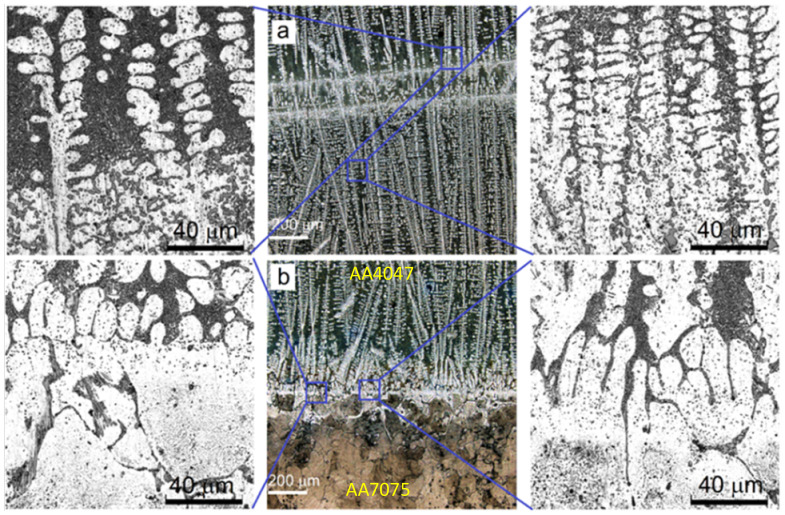
Microstructures in AA4047/AA7075 sample. As-deposited AA4047 zone I (**a**); transition zone (**b**).

**Figure 3 materials-14-06931-f003:**
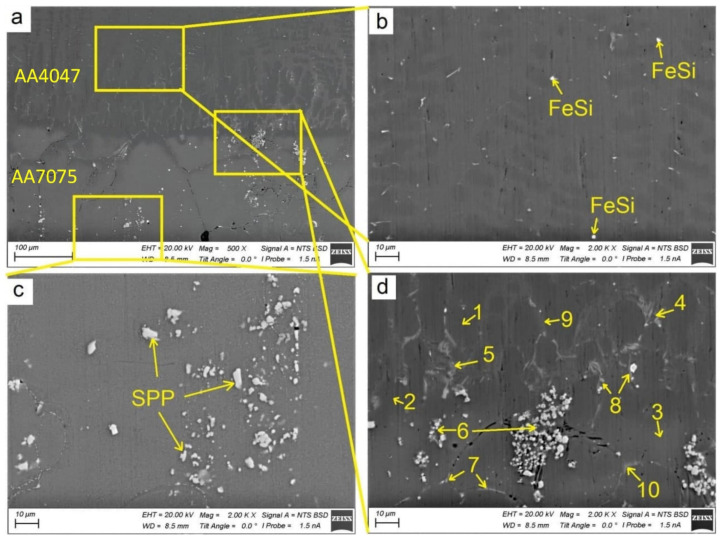
SEM BSE images of Al-Zn-Mg-Cu/Al-Si transition zone (**a**,**d**), Al-1i EBAM part (**b**), AA7075 substrate part (**c**). 1–3—Al-rich, 4–5—Si-rich, 7–8—Si-Mg-Cu-Zn, 9–10—Si-medium. SPP—secondary phase particles.

**Figure 4 materials-14-06931-f004:**
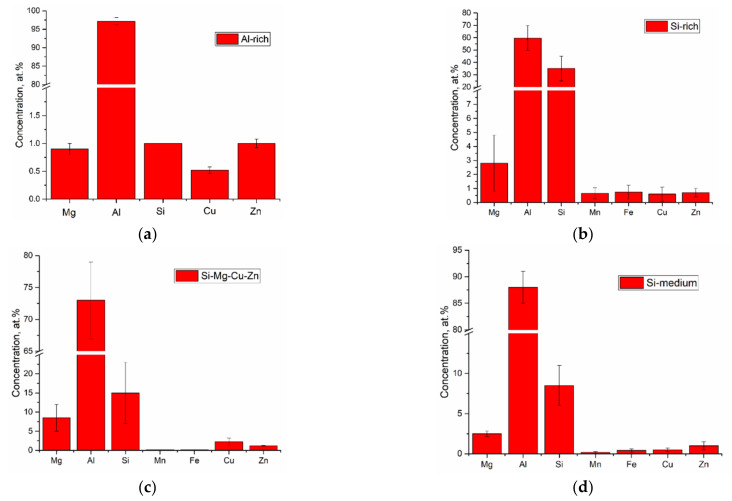
EDS element composition of particles shown in Figure 3. Al-rich (**a**), Si-rich (**b**), Si-Mg-Cu-Zn (**c**), Si-medium (**d**).

**Figure 5 materials-14-06931-f005:**
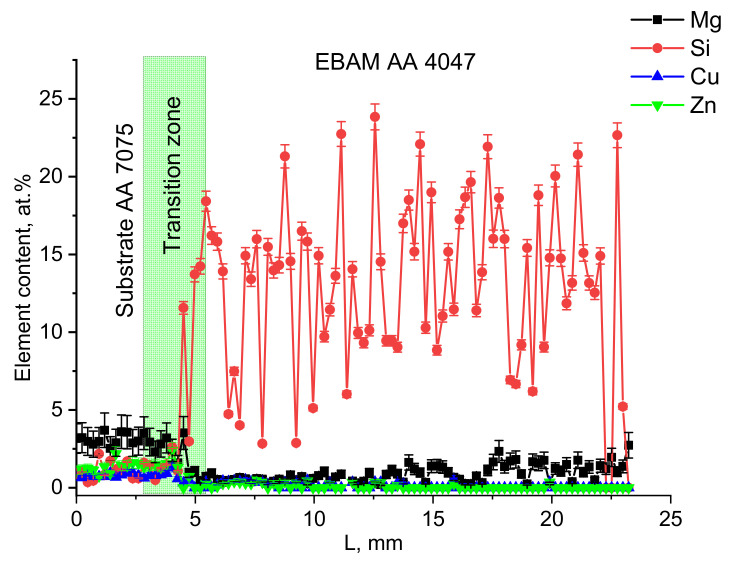
EDS profiles for Mg, Si, Cu and Zn as measured along the wall height.

**Figure 6 materials-14-06931-f006:**
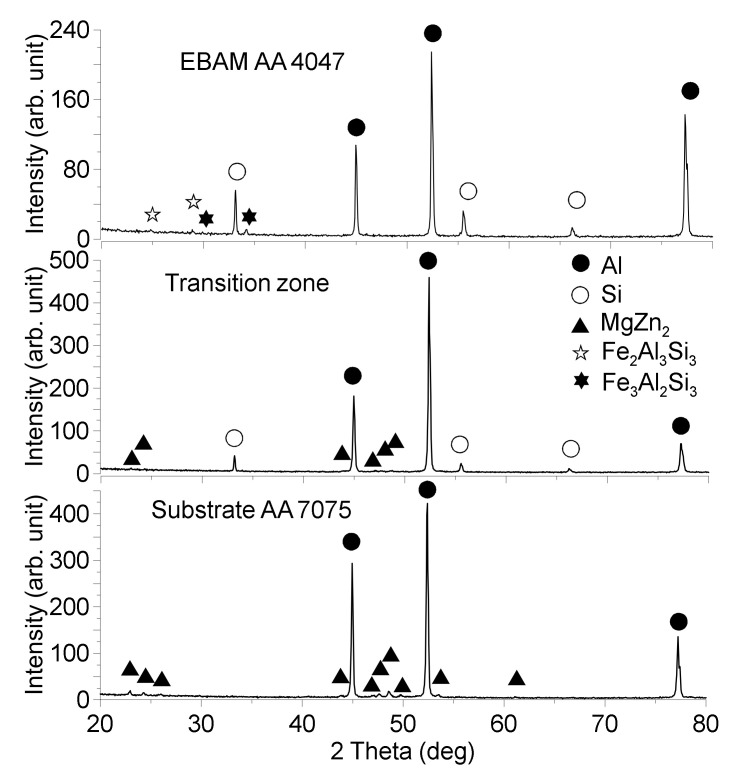
XRD patterns for AA7075 substrate, AA7075/AA4047 transition zone and deposited AA4047.

**Figure 7 materials-14-06931-f007:**
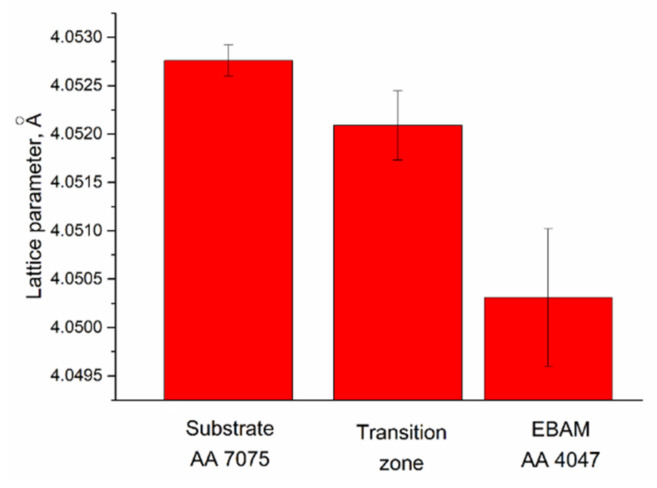
Solid solution lattice parameter as determined for AA7075 substrate, AA7075/AA4047 transition zone and deposited AA4047.

**Figure 8 materials-14-06931-f008:**
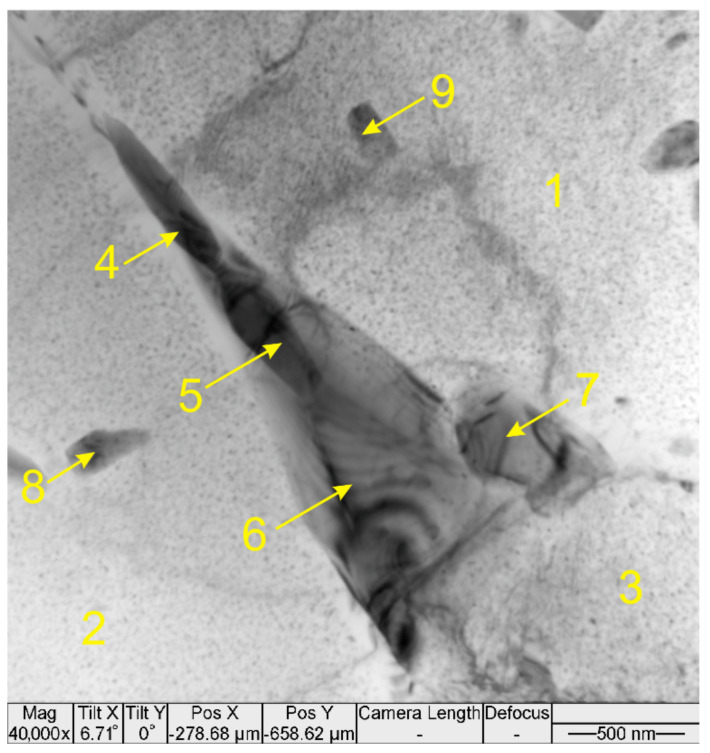
TEM image of a compound ν-phase/S-phase particle in the AA4047/AA7075 transition zone.

**Figure 9 materials-14-06931-f009:**
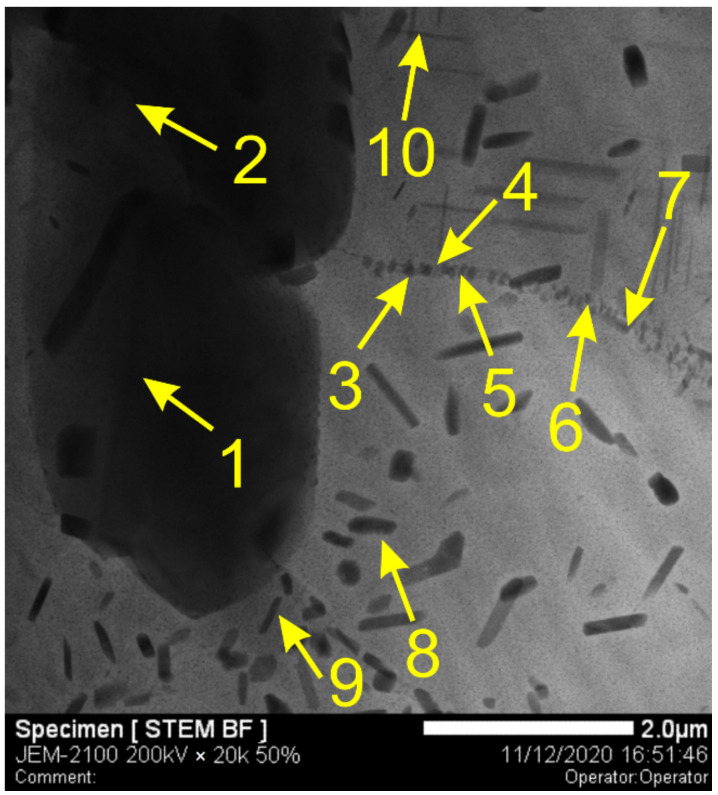
Coarse iron-rich particles and fine Al_2_CuMg and MgZn_2_ precipitates in the AA4047/AA7075 transition zone.

**Figure 10 materials-14-06931-f010:**
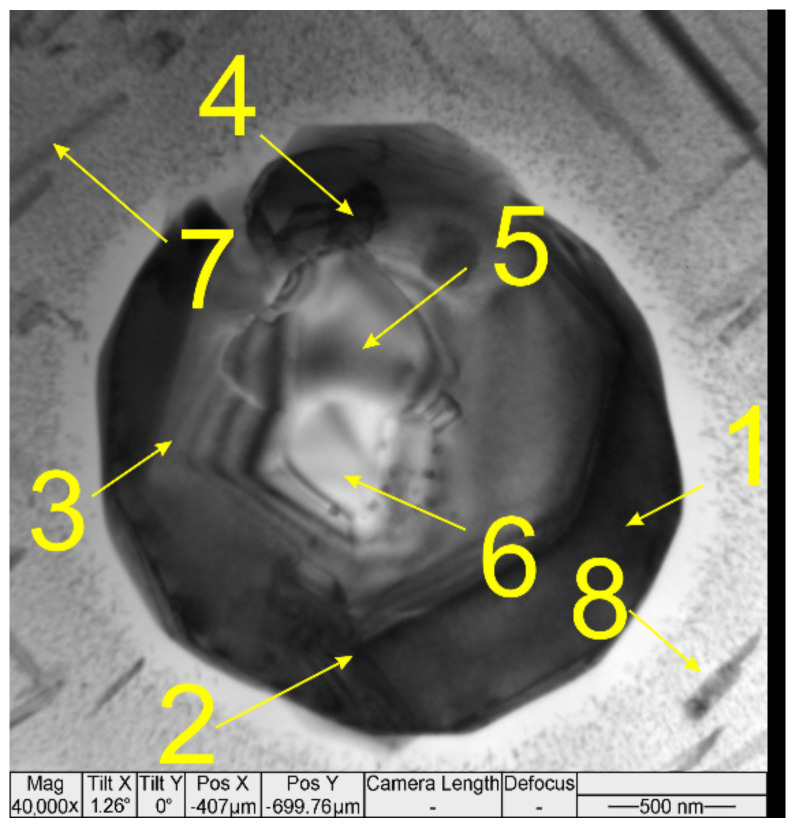
TEM image of a core-shell ν-phase+Mg_2_Si+Al-Zn-Cu-Mg particle in the AA4047/AA7075 transition zone.

**Figure 11 materials-14-06931-f011:**
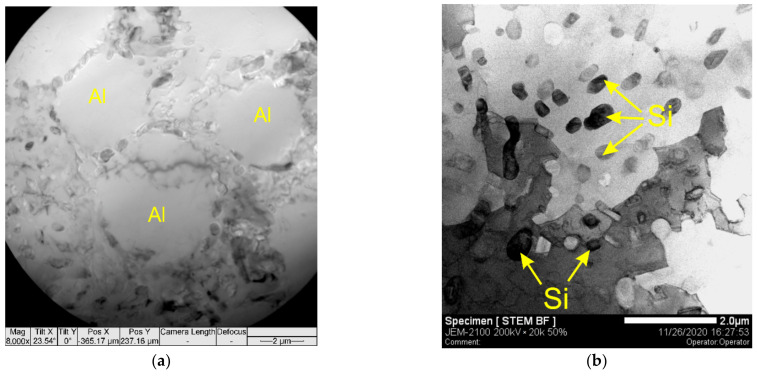
Small α-Al grains with a necklace of Al/Si eutectics (**a**), (**b**) Al/Si eutectic region, (**c**) Si crystallites inside the α-Al grains and (**d**) EDS map of Si distribution corresponding to (**b**).

**Figure 12 materials-14-06931-f012:**
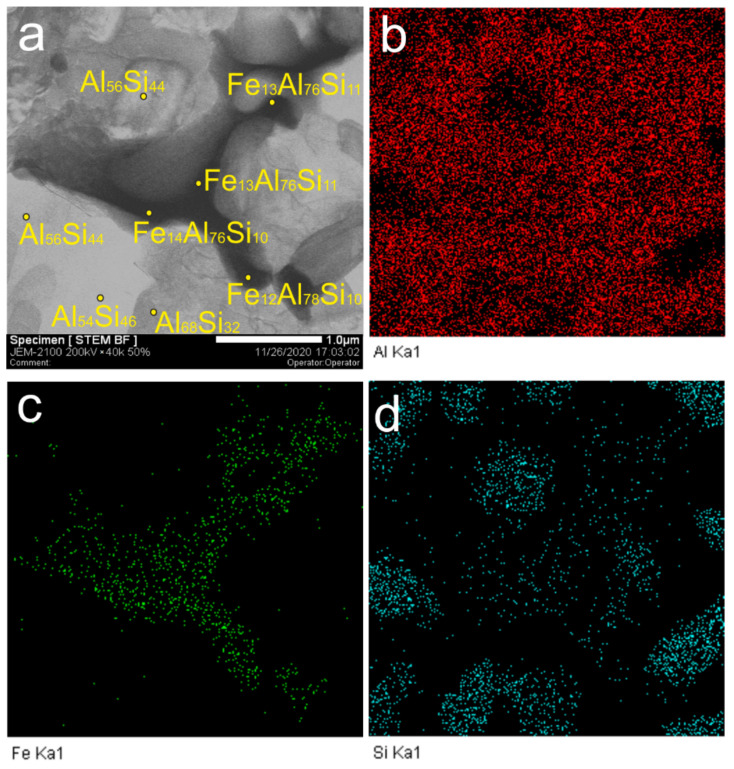
Iron-rich particles (**a**–**c**) and silicon-rich (**a**,**d**) particles.

**Figure 13 materials-14-06931-f013:**
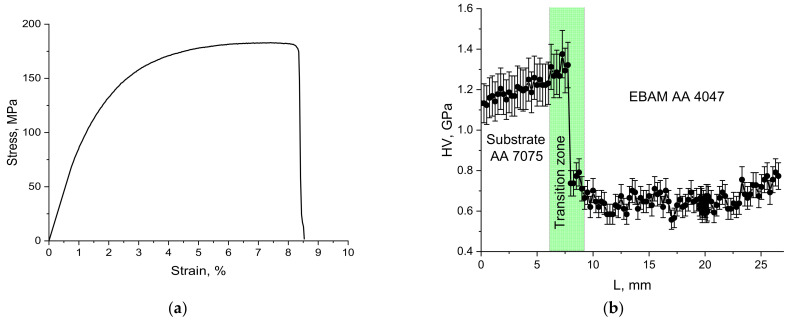
Stress–strain curve for tensile test (**a**) and microhardness profile (**b**).

**Figure 14 materials-14-06931-f014:**
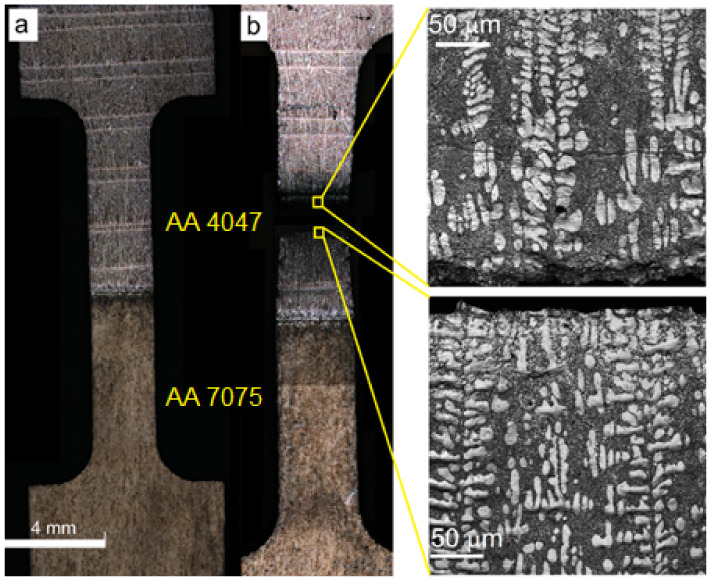
Tensile sample before (**a**) and after (**b**) fracture with dendritic structures in the vicinity of fracture.

**Figure 15 materials-14-06931-f015:**
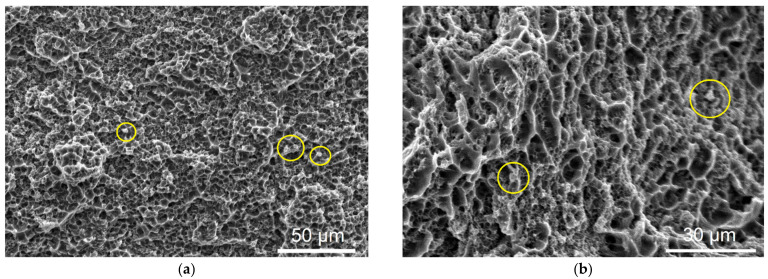
Fracture surface at magnification (**a**) ×500 and (**b**) ×1000.

**Figure 16 materials-14-06931-f016:**
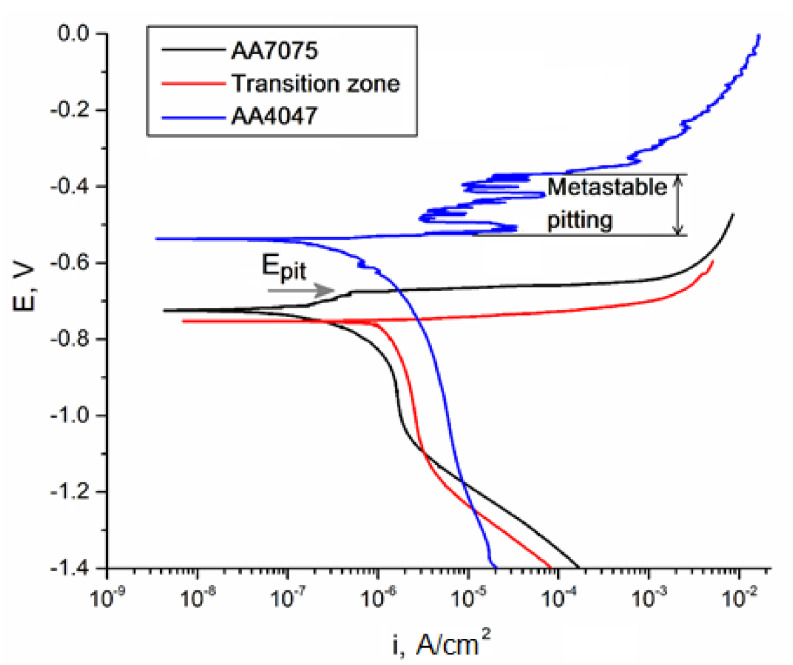
Potentiodynamic polarization curve for studied samples.

**Figure 17 materials-14-06931-f017:**
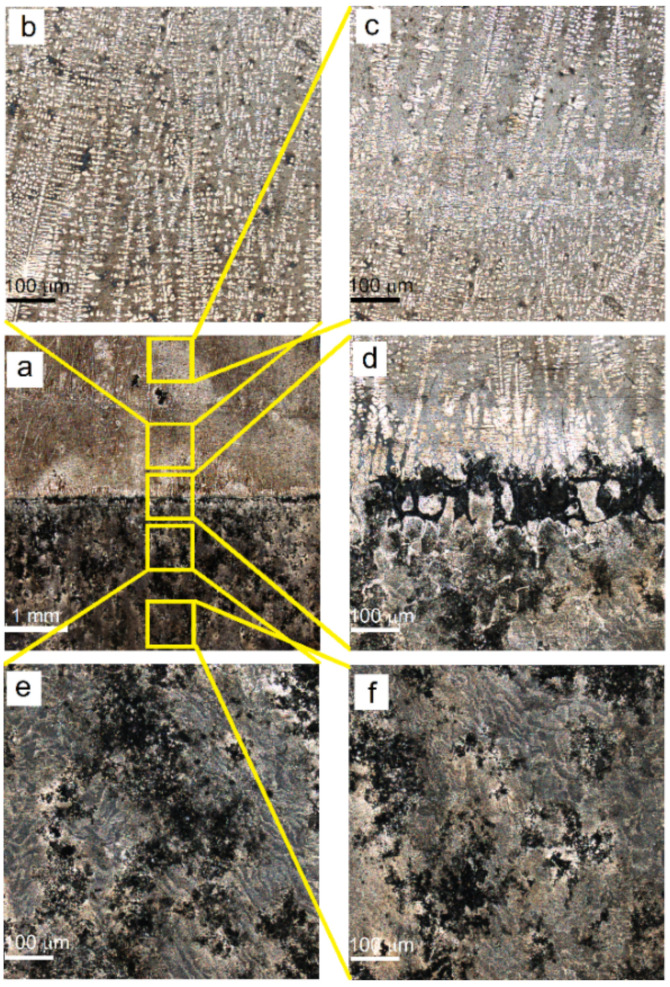
Optical confocal images of corroded surfaces of the samples. Corroded surface macrograph (**a**), as-deposited AA4047 zones (**b**,**c**), transition zone (**d**), heat-affected AA7075 (**e**), as-received AA7075-T42 (**f**).

**Figure 18 materials-14-06931-f018:**
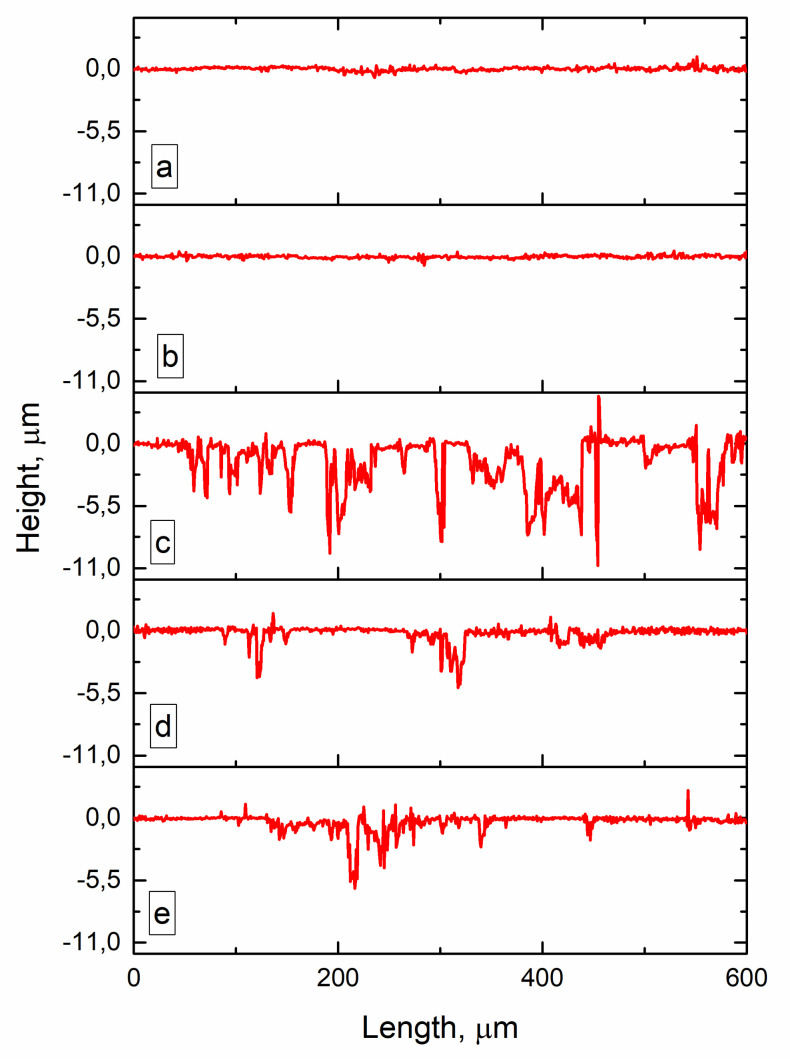
The surface profiles after corrosion according to zones shown in Figure 15. As-deposited AA4047 zones (**a**,**b**), transition zone (**c**), heat-affected AA7075 (**d**), as-received AA7075-T42 (**e**).

**Table 1 materials-14-06931-t001:** Chemical composition of AA4047 wire.

Al	Si	Fe	Cu	Ti
Balance	11.85	0.199	0.006	0.041

**Table 2 materials-14-06931-t002:** Chemical composition of AA7075 substrate [37].

Al	Zn	Cu	Mg	Fe	Si	Mn	Cr	Ti
Balance	5–7	1.4–2	1.8–2.8	0.5	0.5	0.4	0.1	0.05

**Table 3 materials-14-06931-t003:** EDS of the compound particle in Figure 8.

Probe Point	Mg	Al	Cu	Zn	Phase
1–3	1.50 ± 0.08	96.2 ± 0.5	0.60 ± 0.05	1.68 ± 0.05	α-Al
4	21.6 ± 3	56.8 ± 4.5	5.8 ± 1.2	15.9 ± 1.6	ν-phase Mg-Zn-Cu-(Al)
5	31.5 ± 3	36.9 ± 4.2	13.4 ± 2.3	18.1 ± 3.1	
6	21.4 ± 2.5	59.4 ± 6.1	17.9 ± 3	1.3 ± 0.1	S-phase (Al_2_CuMg)
7	12.3 ± 4.0	70.1 ± 8.9	4.5 ± 1.25	13.2 ± 0.4	Al-Zn-Mg-Cu
8	10.6 ± 0.1	79.4 ± 6.0	3.1 ± 1.1	6.9 ± 0.8	Al-Mg-Zn-Cu
9	5.2 ± 0.9	88.2 ± 2.7	2.2 ± 0.4	4.4 ± 0.07	

**Table 4 materials-14-06931-t004:** EDS of particles in Figure 9.

Probe Point	Al	Mn	Mg	Fe	Cu	Zn
1–2	68.7 ± 1.3	4.3 ± 0.9	-	17.9 ± 1.4	6.2 ± 1.8	2.9 ± 0.4
3 and 6	82.9 ± 0.3	-	8.9 ± 0.3	-	6.6 ± 0.2	1.5 ± 0.1
4	75.7 ± 1.1	-	13.8 ± 0.6	-	8.6 ± 0.8	1.8 ± 0.1
5	90.9 ± 2.1	-	4.1 ± 0.8	-	3.1 ± 0.3	1.9 ± 0.2
7	84.7 ± 1.9	-	5.7 ± 0.9	-	3.0 ± 0.3	6.5 ± 1.7
8	74.7 ± 1.6	-	7.9 ± 1.0	-	0.8 ± 0.1	16.6 ± 1.3
9	81.5 ± 1.3	-	5.7 ± 1.3	-	0.5 ± 0.1	12.3 ± 1.0
10	80.2 ± 1.4	-	10.7 ± 1.4	-	8.1 ± 0.9	1.0 ± 0.4

**Table 5 materials-14-06931-t005:** EDS of the core-shell ν-phase/Mg_2_Si/Al-Zn-Cu-Mg particle.

Probe Point	Mg	Al	Si	Cu	Zn
1–4	36.7 ± 6.7	29.9 ± 6.3	-	15.7 ± 2.7	17.1 ± 4.5
5–6	60.6 ± 1.2	2.8 ± 0.3	35.4 ± 0.8	1.2 ± 0.2	-
7	5.5 ± 1.4	87.6 ± 5.9	-	5.7 ± 0.5	1.2 ± 0.4
8	9.2 ± 1.4	75.5 ± 3.6	-	1.54 ± 0.4	13.7 ± 3.4

**Table 6 materials-14-06931-t006:** Polarization parameters for studied samples.

Zone of interest	E_corr_, mV	i_corr_, A/cm^2^
AA4047	−539	7.7 × 10^−8^
Transition zone	−754	1.22 × 10^−6^
AA7075-T42	−725	3.25 × 10^−7^

## Data Availability

The data presented in this study are available on request from the corresponding author.

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
