# Peer review of "Microstructure and Corrosion Resistance of AA4047/AA7075 Transition Zone Formed Using Electron Beam Wire-Feed Additive Manufacturing"

_materials, 2021, doi:10.3390/ma14226931_

Round 1

Reviewer 1 Report

Good and well researched paper. Intresting content especially for AL using companies. Well done

Author Response

Response to Reviewer 1 Comments

Point 1: Good and well researched paper. Interesting content especially for AL using companies. Well done

A: Thank you.

Reviewer 2 Report

In this work, the authors investigated AA4047/AA7075 clad materials manufactured by wire-feed additive manufacturing. Although the authors conducted several experimental works, some parts of manuscript are not so clear that might be confused by readers. Therefore, the reviewer thinks this manuscript is not suitable to publish in Materials in the present form.

(1) Figure 3: If the authors measured EDS, it will be better to represent EDS element distribution maps at the same position. Because some heterogeneous regions are observed in the BSE observation result, element distribution map may be useful to clarify it.

(2) Figure 5: Where the authors measured EDS profiles? Please clarify the measurement line in the SEM image. Moreover, the reviewer concerns about the chemical content deviation in each measurement point. It will be better to mark error bars in each measurement points if possible.

(3) Figures 8-10: Please provide a detailed figure caption. Although the authors mentioned all the images were observed at the transition zone, the authors did not mention the details of precipitates which were not indicated by arrows.

(4) Figure 12: Please add the scale bar to check the precipitate size.

(5) Figure 13(a): How many samples were tested for tensile test? If the authors performed several tensile tests, they need to represent average yield strength, tensile strength and elongation with its deviation.

(6) Figure 13(a): Moreover, the reviewer thinks the elastic constant of present Al alloy looks too low. Therefore, the authors need to clarify how they measured elongation during tensile test.

(7) Figure 13(b): The authors need to add error bars in each hardness measurement points.

(8) Figure 15: The reviewer cannot understand what the authors want to discuss with this fracture surface. The reviewer thinks the precipitates may induce cracks during tensile deformation and this might be observed in the fracture surface. Therefore, the reviewer recommends to re-check the fracture surface to clarify it.

(9) Figure 18: The reviewer found some typos in the manuscript. For example, Figure 15 in the caption of “The surface profiles after corrosion according to zones shown in Figure 15” should be Figure 17.

Author Response

In this work, the authors investigated AA4047/AA7075 clad materials manufactured by wire-feed additive manufacturing. Although the authors conducted several experimental works, some parts of manuscript are not so clear that might be confused by readers. Therefore, the reviewer thinks this manuscript is not suitable to publish in Materials in the present form.

Point 1: Figure 3: If the authors measured EDS, it will be better to represent EDS element distribution maps at the same position. Because some heterogeneous regions are observed in the BSE observation result, element distribution map may be useful to clarify it.

A: The EDS maps were not representable enough to show the element distribution in this case since the differences between these aluminum-base sorts of particles were mainly quantitative and therefore it was important to determine their composition. Also the EDS maps usually take too much space in the manuscript but allow hardly to identify the precipitates. 

Point 2: Figure 5: Where the authors measured EDS profiles? Please clarify the measurement line in the SEM image. Moreover, the reviewer concerns about the chemical content deviation in each measurement point. It will be better to mark error bars in each measurement points if possible.

A: The EDS profiles in Fig.5 were obtained along the sample growth direction starting from a substrate up to the built-up metal top 25 mm away from the substrate. No SEM image was made with the magnification low enough to show this line. The EDS element error bars were induicated in the Fig.5 new version.

Point 3: Figures 8-10: Please provide a detailed figure caption. Although the authors mentioned all the images were observed at the transition zone, the authors did not mention the details of precipitates which were not indicated by arrows.

A: Captions to Fig.8-10 was revised so that fine particles were identified in Tables 3-5.

Point 4: Figure 12: Please add the scale bar to check the precipitate size.

A: Thank you. Corrected.

Point 5: Figure 13(a): How many samples were tested for tensile test? If the authors performed several tensile tests, they need to represent average yield strength, tensile strength and elongation with its deviation.

A: Total 6 samples were tested for tensile strength. Fig.13a shows a typical stress-strain diagram. Mean values of yield stress, ultimate stress and elongation-to-fracture were shown in the text. 

Point 6: Figure 13(a): Moreover, the reviewer thinks the elastic constant of present Al alloy looks too low. Therefore, the authors need to clarify how they measured elongation during tensile test

A: The relative elongation was determined according to a formula below:

δ=((lf - ls)/ls)*100%

where ls – initial sample gauge length measured at the room temperature, mm; lf  is the sample gauge length after the fracture, mm. How else could it have been done? Let us remind also that the as-cast structure composed of columnar grains can show more ductility as compared to that of equiaxed grained ones.

Point 7: Figure 13(b): The authors need to add error bars in each hardness measurement points.

A: Thank you. Corrected

Point 8: Figure 15: The reviewer cannot understand what the authors want to discuss with this fracture surface. The reviewer thinks the precipitates may induce cracks during tensile deformation and this might be observed in the fracture surface. Therefore, the reviewer recommends to re-check the fracture surface to clarify it.

A: The fracture occurred in the as-deposited AA4047 zone where there are no components other than Al-dendrites, excess Si-crystallites and Al/Si eutectics. It could be that Si-crystallites served as stress concentrators but this as-cast structure was heterogeneous enough by itself.

Point 9: Figure 18: The reviewer found some typos in the manuscript. For example, Figure 15 in the caption of “The surface profiles after corrosion according to zones shown in Figure 15” should be Figure 17.

A: Thank you. Corrected.

Reviewer 3 Report

The paper is interesting, relevant and could be suitable for publication in the Materials  after some major revision

  1. The introduction required some revision and update with paper publish with similar work regards the interface between wrought and AM materials, for example:
    1. Kohler et al. “wire and arc additive manufacturing of Aluminum components”, Metals 2019, 9, 608
    2. Dolev et al. “Ti-6Al-4V hybrid structure mechanical properties—Wrought and additive manufactured powder-bed material”, Additive Manufacturing, 37, 2021, 101657
  2. AA4047 is a common wire for welding and is expected to be used in welding, explain the reason why using this particular wire!
  3. In section 2: Materials and Methods – add the process information, ie. Wire feed, the thickness of the built layer, vacuum level
  4. Due to the vacuum environments, the heat is developed in the process, add the information on the cooling rate (if you used a cooling system) and what was the temperature prior to exposing the aluminum to an air environment.
  5. The samples present in figure 2a are very small, add the dimension of the tensile samples (on a figure or separate drawing).
  6. Figures 2 and 3, mark on figures the material on each side (it should be clear which is the AA7075 and which is AA4047)
  7. Page 6 figure 5: the high scattering of the Si in the AA4047 built layers indicate that process is not stable !!! in this condition the reputability of the experiments and the quality of the results are questionable
  8. In addition to remark 7, In EB welding the Si tends to vaporize, the initial SI content is 11.85 (according to table 1), it’s required a very good explanation of how the authors got more than 20% after the deposition process !!!
  9. Section 3.4 page 12:  what is the reputability of the results presented in figure 13?  Add the statistical data of tests and error bar on figures.
  10. Figure 14: see my remark 6
  11. Figure 15: the magnification scale is not clear on figures
  12. Section 5 – last sentence in the conclusions “These results provide …..”, This conclusion is not relevant and it is not clear how the authors rich to “an optimal wire feed electron beam AM of functionally graded materials”  the paper presents results and not optimization of the process  !!!

Author Response

Response to Reviewer 3 Comments

The paper is interesting, relevant and could be suitable for publication in the Materials after some major revision.

Point 1: The introduction required some revision and update with paper publish with similar work regards the interface between wrought and AM materials, for example:

  1. Kohler et al. “wire and arc additive manufacturing of Aluminum components”, Metals 2019, 9, 608
  2. Dolev et al. “Ti-6Al-4V hybrid structure mechanical properties—Wrought and additive manufactured powder-bed material”, Additive Manufacturing, 37, 2021, 101657

A: Thank you. Corrected.

Point 2: AA4047 is a common wire for welding and is expected to be used in welding, explain the reason why using this particular wire!

A: AA4047 has good corrosion resistance and therefore could be used for surfacing the less corrosion resistant 7XXX alloys. The fact that it accessible as a welding wire  served as its advantage in case of using the electron beam additive manufacturing for obtaining resilient and corrosion resistant combined dissimilar alloy components.  

Point 3: In section 2: Materials and Methods – add the process information, i.e. Wire feed, the thickness of the built layer, vacuum level

A:  Residual pressure in the chamber was - 5·10-5 Pa.  Deposition rate, wire feed rate, beam current and accelerating voltage were 380 mm/min, 1344 mm/min, 20 μA and 30kV, respectively. The wall dimensions (with substrate) were 95 mm, 39 mm and 5 mm.

Corresponding information was added to the manuscript

Point 4: Due to the vacuum environments, the heat is developed in the process, add the information on the cooling rate (if you used a cooling system) and what was the temperature prior to exposing the aluminum to an air environment.

A: Cooling rate was not measured. Water cooling table was used with water temperature 13-16°C. Samples was exposed to air after full cooling to the room temperatures. 

Point 5: The samples present in figure 2a are very small, add the dimension of the tensile samples (on a figure or separate drawing).

A: Thank you. The sample dimensions were indicated in Fig.1 as follows:

Point 6: Figures 2 and 3, mark on figures the material on each side (it should be clear which is the AA7075 and which is AA4047)

A: Corrected.

Point 7: Page 6 figure 5: the high scattering of the Si in the AA4047 built layers indicate that process is not stable !!! in this condition the reputability of the experiments and the quality of the results are questionable

A: I have no idea what kind of process is implied here. The presence of dendrites is the evidence that solidification rate was low enough. Dendrite liquation is the reason for both structural and element distribution inhomogeneity. The fact that the as-deposited AA4047 is structurally inhomogeneous becomes clear from the very beginning of the study, see Figure 2. Along with Al-dendrites there are Al/Si eutectics as well as isolated Si crystallites. Therefore, nobody can expect smooth point-by-point concentration profile for Si. Even integral concentrations in the Figure below shows Si concentrations higher that the standard mean values from tables.

Point 8: In addition to remark 7, In EB welding the Si tends to vaporize, the initial SI content is 11.85 (according to table 1), it’s required a very good explanation of how the authors got more than 20% after the deposition process !!!

A: The answer to this comment has already been given. Of course no somewhat noticeable evaporation of silicon could happen during the EBAM in contrary to the well-known evaporation of low-boiling point elements as Mg and Zn. It is not surprising since the evaporation point for Si is at 2335°C, i.e. almost as high as that for aluminum (2735°C). For comparison the boiling points of magnesium and zinc are 1090°C and 907°C. Also, nothing surprising is in detecting high local concentrations of Si in extremely inhomogeneous as-cast structures. Please, pay attention that as-cast structures always show high scatter in element concentrations. 

Point 9: Section 3.4 page 12:  what is the reputability of the results presented in figure 13?  Add the statistical data of tests and error bar on figures.

A: Thank you. Corrected. Fig.13a represents typical stress-strain curve while mean strength characteristics were shown in the text above.

Point 10: Figure 14: see my remark 6

A: Thank you. Corrected.

Point 11: Figure 15: the magnification scale is not clear on figures

A: Thank you. Corrected.

Point 12: Section 5 – last sentence in the conclusions “These results provide …..”, This conclusion is not relevant and it is not clear how the authors rich to “an optimal wire feed electron beam AM of functionally graded materials”  the paper presents results and not optimization of the process  !!!

A: This sentence was eliminated

Round 2

Reviewer 2 Report

The authors responses are appropriate to solve the reviewer's concern. The present manuscript looks suitable to publish in Materials.

Reviewer 3 Report

I ACCEPT THE PAPER FOR PUBLICATION.  THE AUTHORS CORRECT ALL REMARKS. 

PLEASE NOTE THAT ONE OF THE FIGURES INCLUDE A RUSSIAN LANGUAGE